# Global Distribution of *mcr* Gene Variants in 214K Metagenomic Samples

Hannah-Marie Martiny,ᵃ Patrick Munk,ᵃ Christian Brinch,ᵃ Judit Szarvas,ᵃ Frank M. Aarestrup,ᵃ Thomas Nordahl Petersenᵃ

ᵃResearch Group for Genomic Epidemiology, Technical University of Denmark, Kgs. Lyngby, Denmark

**ABSTRACT** Since the initial discovery of a mobilized colistin resistance gene (*mcr-1*), several other variants have been reported, some of which might have circulated a while beforehand. Publicly available metagenomic data provide an opportunity to reanalyze samples to understand the evolutionary history of recently discovered antimicrobial resistance genes (ARGs). Here, we present a large-scale metagenomic study of 442 Tbp of sequencing reads from 214,095 samples to describe the dissemination and emergence of nine *mcr* gene variants (*mcr-1* to *mcr-9*). Our results show that the dissemination of each variant is not uniform. Instead, the source and location play a role in the spread. However, the genomic context and the genes themselves remain primarily unchanged. We report evidence of new subvariants occurring in specific environments, such as a highly prevalent and new variant of *mcr-9*. This work emphasizes the importance of sharing genomic data for the surveillance of ARGs in our understanding of antimicrobial resistance.

**IMPORTANCE** The ever-growing collection of metagenomic samples available in public data repositories has the potential to reveal new details on the emergence and dissemination of mobilized colistin resistance genes. Our analysis of metagenomes deposited online in the last 10 years shows that the environmental distribution of *mcr* gene variants depends on sampling source and location, possibly leading to the emergence of new variants, although the contig on which the *mcr* genes were found remained consistent.

**KEYWORDS** antimicrobial resistance, metagenomics, microbiome

Antimicrobial resistance (AMR) is considered one of the most significant threats against human and animal health (1). Over the years, we have observed the emergence of a multitude of novel antimicrobial resistance genes (ARGs), and it is generally believed that such genes have emerged and evolved in the commensal flora for a long time prior to being detected in pathogenic isolates (2).

Colistin is an important antibiotic used as a last-resort choice to treat multidrug-resistant (MDR) and carbapenem-resistant bacteria (3). Before 2015, colistin resistance was believed to be only due to mutational and regulatory changes in chromosomal genes. A mobilized colistin resistance gene, *mcr-1*, was discovered in 2015 on a plasmid in *Escherichia coli* isolates from China (4), raising concern in the scientific community about the possibility of resistance spreading more rapidly by horizontal gene transfer by mobile genetic elements (MGEs) (4, 5). Immediately following the first report, a large number of studies were initiated in several countries around the world, and it was soon determined that *mcr-1* was already widespread and has now been detected in all continents (6–8). In initial reports, the most frequent isolates were sampled from livestock sources, followed by humans, meat, and food products (9). Since then, several new variants of *mcr* genes have also been identified, named *mcr-2* to *mcr-10* and sharing 81%, 32.5%, 34%, 36%, 83%, 35%, 31%, 36%, and 29.31% amino acid sequence

Address correspondence to Hannah-Marie Martiny, hanmar@food.dtu.dk.

The authors declare no conflict of interest.

identity to *mcr-1*, respectively (10–17). Retrospective screening of bacterial isolates and available sequences of mainly pathogenic isolates showed a more widespread occurrence and prior evolution of *mcr* before its initial discovery (8, 18, 19).

However, investigating only pathogenic strains or cultivable bacteria will only provide limited insight into the potential reservoirs of such novel ARGs. As documented by our research group and others, investigating the entire microbiome provides additional information on the presence and diversity of ARGs (7, 20–22). Today most researchers conducting microbiome studies are depositing their raw data in global repositories, allowing other researchers to reanalyze the data and provide novel insight.

This study was conducted to investigate the occurrence and global dissemination of known *mcr* gene variants in publicly available metagenomic data sets. We downloaded 442 Tbp of raw reads from 214,095 metagenomic data sets and determined the presence and abundance of 9 *mcr* gene variants. We found that only a small subset of the metagenomic data sets was positive for at least one of the *mcr* genes but that the abundance gradually increased as a function of time. The distribution of each variant varied by region and sampling source, but the genomic background of each gene was consistent across different environments. However, several subvariants are observed with conserved single nucleotide polymorphisms (SNPs) across multiple samples. Despite the sparsity of the data once stratified by the presence of *mcr* genes, our analysis suggests that multiple factors have likely influenced the dissemination of colistin resistance and that screening publicly available metagenomic samples can, together with single isolates, further deepen our understanding of the distribution of *mcr* gene variants.

## RESULTS

**Data set.** After retrieval, quality checking, and trimming of the raw sequencing reads of the 214,095 metagenomic data sets, we aligned the reads against ARGs and 16S rRNA sequences using the assembly-free method KMA. The resulting counts of read fragments aligned to different reference sequences were used to analyze the distribution and abundance of *mcr* genes. The abundance of an *mcr* gene was calculated as the fragment count of that gene over the total amount of bacterial fragments for a sample or a group, whereas the fragment count for ARGs was the only one used for statistical analyses.

Out of the 214,095 metagenomes, we found that 2.09% (4,465) of them contained read fragments aligning to at least 1 of the 9 *mcr* gene variants. The average number of reads per *mcr*-positive sample was 27 million reads, and on average, 0.003% of the reads were aligned to *mcr* genes. Among the variants in the *mcr* family, *mcr-1* and *mcr-9* were the most frequent, with 25.91% and 57.47% of the *mcr*-aligned reads aligning to these variants, and disseminated across 10 and 13 sampling years, 21 and 56 countries, and 23 and 61 hosts, respectively. The rarest variants were *mcr-2*, *mcr-6*, and *mcr-8* with read frequencies of 0.03%, 0.01%, and 0.08%, respectively, and their metagenomic origins were more restricted (Table 1). Overall, different log-ratio abundance levels seemed to be different across the sampling years in different countries and hosts (Fig. S1).

**Level of *mcr* variants over time.** The *mcr*-positive metagenomic samples were collected between 2003 and 2019, with the exception of 2005, in which no *mcr* fragments were detected (Fig. 1). Only two metagenomes sampled in 2003 contained *mcr* fragments, and a single *mcr*-positive metagenome was from 2004. Onward, the percentage of positive samples fluctuated, with the lowest value of 0.5% in 2008 and the highest of 6.4% in 2019 (Fig. S1). All the variants were frequently found in samples from 2016 to 2017, except *mcr-6*, which was only found in 2012 (Table 1).

We found that the log ratio abundance of aligned read fragments fluctuated for the nine variants in each sampling year (Fig. 1). The oldest positive metagenomes were sampled in 2003 and 2004 and contained only *mcr-3* and *mcr-5*. From 2006, the other variants began to emerge. *mcr-1* was detected first in 2009 at a low log abundance, and increased levels were observed between 2011 and 2019. Similarly, *mcr-9* could be detected in small amounts in metagenomes from 2007. In 2012 and 2013, *mcr-9* was

**TABLE 1** Read alignment of each *mcr* variant across different sample types

| | *mcr-1* | *mcr-2* | *mcr-3* | *mcr-4* | *mcr-5* | *mcr-6* | *mcr-7* | *mcr-8* | *mcr-9* |
|---|---|---|---|---|---|---|---|---|---|
| Origin[a] | | | | | | | | | |
| Read frequency (%) | 25.91 | 0.03 | 10.33 | 0.98 | 1.86 | 0.01 | 3.32 | 0.08 | 57.47 |
| No. of yrs | 10 | 6 | 14 | 13 | 13 | 1 | 13 | 11 | 13 |
| No. of countries | 21 | 6 | 59 | 42 | 43 | 1 | 27 | 14 | 56 |
| No. of hosts/reservoirs | 22 | 6 | 49 | 54 | 40 | 1 | 43 | 8 | 60 |
| Yr[b] | | | | | | | | | |
| 2010 | | 16.67 | | | | | | | |
| 2012 | | | | | 9.03 | 100.00 | | 20.00 | |
| 2013 | | | | 17.04 | | | | | |
| 2014 | | | | | | | | | 23.89 |
| 2015 | 37.66 | | | | | | | | |
| 2016 | | | 58.77 | 20.89 | 61.11 | | 30.68 | 16.00 | 22.11 |
| 2017 | 22.86 | 16.67 | 14.36 | | | | 17.40 | | |
| Country[c] | | | | | | | | | |
| Angola | | 16.67 | | | | | | | |
| Cambodia | 6.36 | | | | | | | | |
| China | 68.96 | | | | | | 21.9 | | |
| Denmark | | | 40.11 | 12.16 | 36.24 | | | | |
| France | | | | | | 100.0 | | | |
| Kenya | | | | | | | | 11.10 | |
| Netherlands | | 16.67 | | | | | | | |
| USA | | | 13.43 | 39.38 | 18.78 | | 23.63 | 11.11 | 45.18 |
| Host/reservoir[d] | | | | | | | | | |
| *Homo sapiens* | 32.51 | 33.33 | | 23.71 | | | | 19.35 | 56.46 |
| Panda | 22.59 | | | | | | | | |
| Activated sludge metagenome | | | | | 5.93 | | | | |
| Freshwater metagenome | | | 10.16 | | | | 18.24 | | |
| Marine metagenome | | 22.22 | | | | | | 48.39 | |
| Microbial mat metagenome | | | | | | 100.0 | | | |
| Wastewater metagenome | | | 65.77 | 26.20 | 60.81 | | 25.88 | | 13.44 |

[a]Read frequencies and counts of unique sampling origins, i.e., the number of years, countries, and hosts/reservoirs.
[b]The top two sampling years for the given variant was the most abundant in abundant in is shown in percentage of *mcr*-mapped reads.
[c]The top two sampling countries as described in footnote b.
[d]The top two sampling hosts and reservoirs as described in footnote b.

the most abundant variant, with 81% and 86% of the read fragments aligning to this gene. In 2007, only 3% of the *mcr* read fragments aligned to *mcr-7*, but more and more fragments for each year were assigned to the *mcr-7* gene and peaked in 2019 with 95% of the *mcr* fragments aligned being to it.

Significant levels of different *mcr* genes were observed for sampling years 2011, 2013, 2014, 2015, 2016, and 2017 ($P$ value $<$ 0.05, Fig. 4a). Even though the variance of *mcr* levels within the sampling years was high, several variants stood out as having higher or lower levels in specific years compared to other years. In 2011, *mcr-3* had a higher abundance than expected and continued to be high in 2013 to 2014, together with *mcr-1* and *mcr-5*. *mcr-9* was lower in those years. In 2016, the metagenomic picture changed as *mcr-3* and *mcr-5* had decreased levels, while *mcr-1*, *mcr-4*, *mcr-7*, and *mcr-9* were increased.

**Geographical distribution of *mcr* gene variants.** The 9 variants were spread across 95 different sampling locations (Fig. S1), although samples from different world regions were often different in which variant they were positive for (Table 1). A higher abundance of *mcr* gene variants was observed in the Americas, Asia, and Europe, and decreased abundances were observed in Africa. The highest total log-ratio abundances of *mcr* fragments could be found in metagenomes from Australia, Lake Huron (USA), and Cambodia, and lowest levels, in Kiribati, Greece, and the Caribbean Sea (Fig. 2).

The individual variants were not equally distributed worldwide; instead, it seemed like specific variants were restricted to specific regions (Fig. 2). For example, the variant *mcr-1* was less widely spread worldwide (Americas, Asia, and Europe) than *mcr-9*

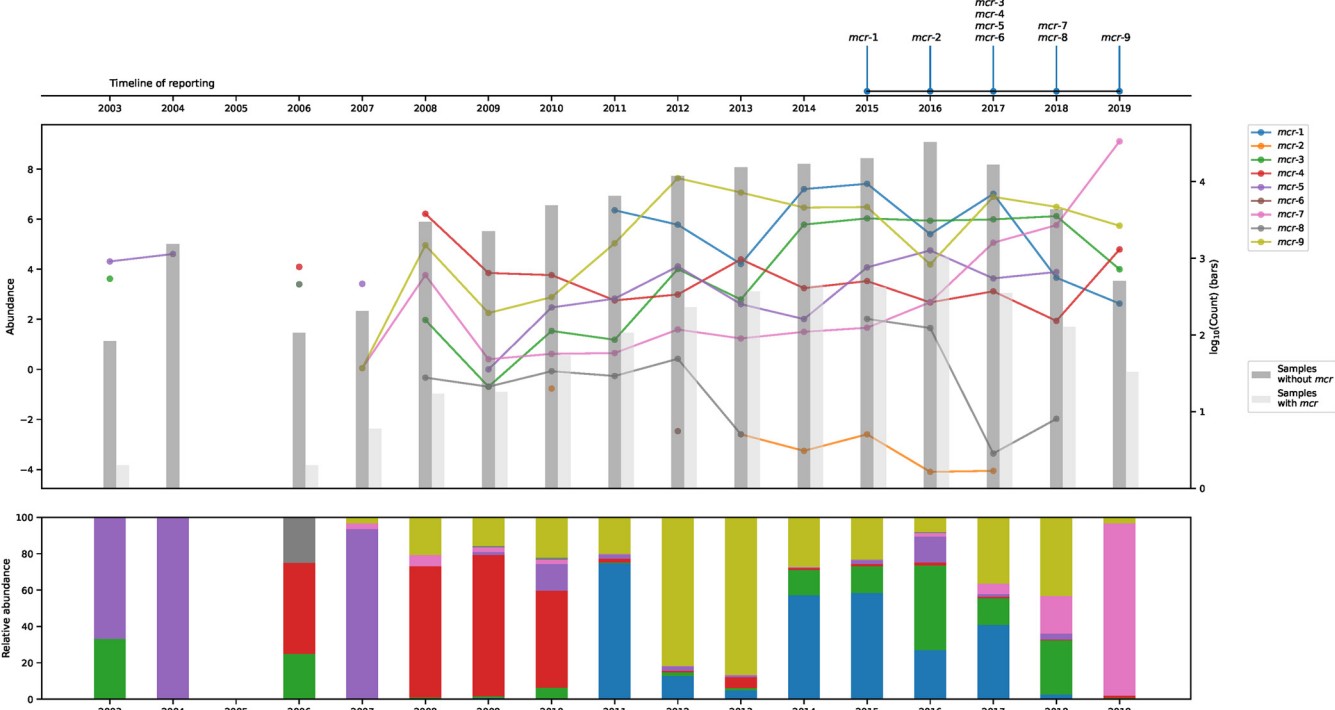

**FIG 1** Discovery and the change of *mcr* genes over time. (Top) Timeline showing when each gene was first reported in the literature. (Middle) Changes in log abundance of aligned *mcr* read fragments over time for each gene are shown, as well as the number of samples with or without an *mcr* hit from each year as bars. (Bottom) The frequency of each gene compared to the total *mcr* amount. Data were normalized with gene lengths to generate the charts.

(Africa, Americas, Asia, Europe, Oceania, Atlantic Ocean, and the Pacific Ocean). No metagenomic locations contained all types of variants. In the Australian metagenomes, *mcr-9* was the most dominant gene, whereas *mcr-4* had high abundance levels in Lake Huron (USA), and *mcr-1* and *mcr-9* had high abundance levels in Cambodian metagenomes. The only location of *mcr-6* was France.

Of the 95 sampling locations, 15 had significant abundances of at least one of the *mcr* variants (*P* value < 0.05); however, the variance in the samples from most of the locations was high and did not have a large effect size compared to other locations (Fig. 4c). Metagenomic locations that showed consistency within the group and were found to be different from the rest of the locations had lower levels of single variants —*mcr-1* in Bulgaria, *mcr-3* in Iceland, *mcr-5* in Malaysia, and *mcr-9* in Cambodia.

**Host- and reservoir-specific *mcr* abundances.** We found *mcr* genes present in 125 different sampling hosts and reservoirs, but with the various variants having different log-ratio fragment abundances (Fig. S2) and the two most frequent types differing for each variant (Table 1). All 6 metagenomes from *Pomacea canaliculate* (golden apple snail) and the 11 *Danio rerio* (zebrafish) metagenomes contained *mcr* fragments. For two of the largest sampling groups, we found 897 out of 1,803 (49.75%) wastewater metagenomes and 13,831 out of 102,211 (1.35%) human-derived samples to be *mcr* positive.

Out of the 125 hosts, only 20 of them showed significant levels of *mcr* gene fragments. These all had higher levels of colistin resistance genes (Fig. 3). The dispersion within most of the 20 hosts was high, and their log-ratio levels were not significantly different from those of the other hosts, except a few (Fig. 4e). The zebrafish samples had lower levels of *mcr-7* than expected, whereas golden apple snail metagenomes had higher levels. Panda metagenomes had elevated levels of *mcr-9* but had slightly smaller amounts of *mcr-3*, *mcr-4*, and *mcr-5*. Metagenomes from pigs (*Sus scrofa* and pig gut) had increased levels of both *mcr-1* and *mcr-9*. Human metagenomes did not have large effect sizes but contained slightly less *mcr-1* and *mcr-9* than expected.

**Diversity of *mcr*-positive metagenomes.** By performing compositional PCA analysis on CLR values, we can visualize the variance in *mcr* read proportions in biplots

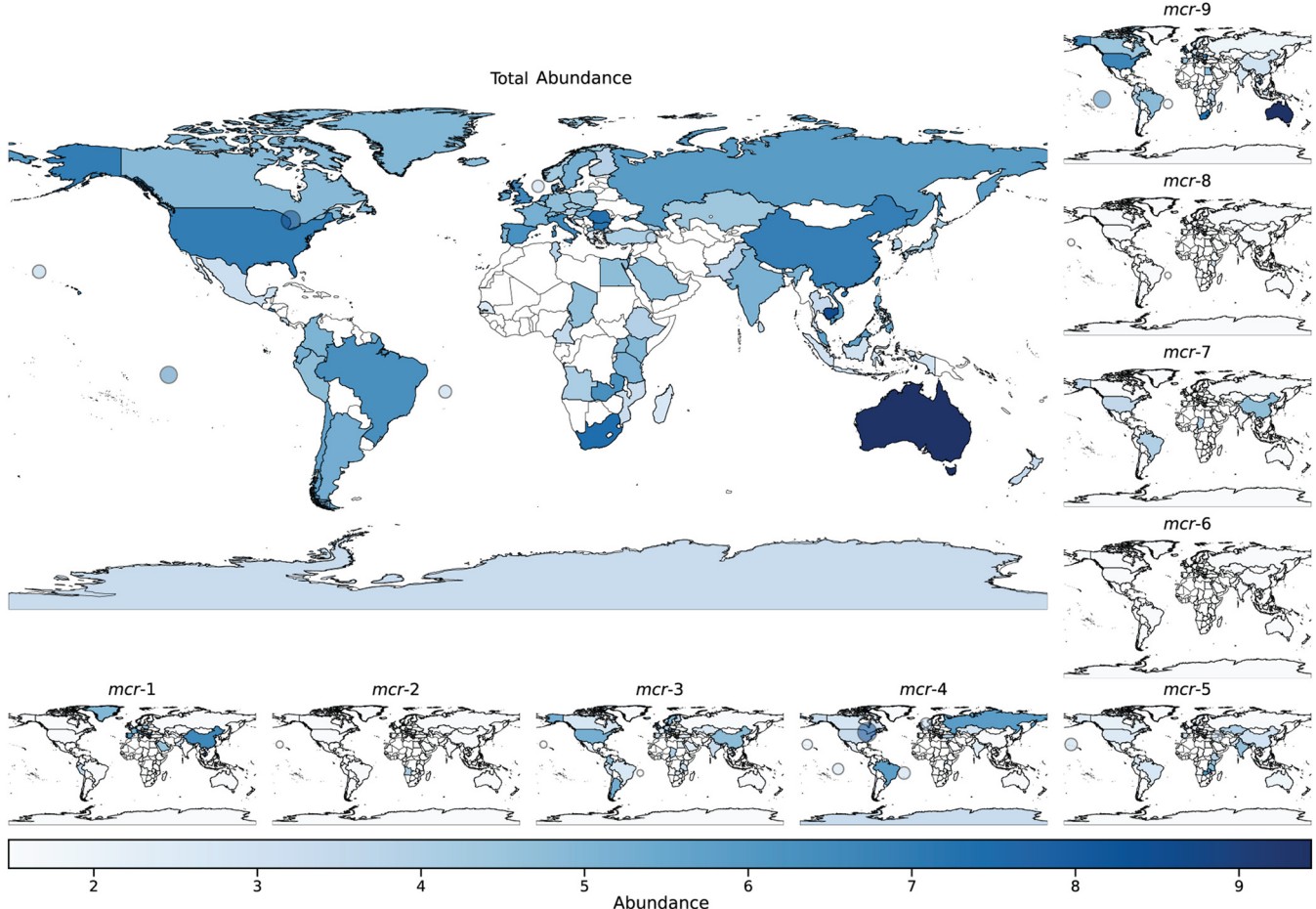

**FIG 2** Global levels of *mcr* genes. The large map shows the total log abundance levels of all *mcr* genes, whereas the nine smaller plots show the individual gene log-scaled abundances worldwide. A circle represents a collection of samples from water containing *mcr* genes. White color indicates an absence of results, not that a specific location does not have any *mcr* genes. The circle markers illustrate water environments.

showing which type of samples make the level of a resistance gene significant (Fig. 4). The biplots highlight a clear separation of metagenomes that contain *mcr-1* or *mcr-9* and show that these samples also differ a lot from each other. None of the samples from the different years are similar, which means that high levels of one of the variants cannot be explained simply due to a specific collection year (Fig. 4b). Instead, we can see that several Panda metagenomes came from China in 2016, which most likely contributed to the higher levels of *mcr-1* in 2016 (Fig. 4d). Likewise, human metagenomes clearly show a geographical separation mainly driven by *mcr-1* being abundant in China and *mcr-9* in the United States and Australia (Fig. 4f, Fig. S3), which could explain that even if these metagenomes contain significant levels of *mcr* genes, we could not observe large effect sizes. Excluding these two most abundant genes suggests, however, that the differences are mainly driven by source and not by year or geographical location (Fig. S4).

**Distribution of *mcr* variants in pathogenic bacterial genomes.** As several studies have performed retrospective screening of pathogenic bacterial isolates, we decided to compare the metagenomic *mcr* abundances to the prevalence in pathogenic single isolates. Out of 912,469 isolates screened by the NCBI Pathogen Detection Pipeline, only 7,934 (0.87%) were shown to carry at least 1 of the *mcr* genes. The majority of the *mcr*-positive isolates contained either *mcr-1* (51.08%) or *mcr-9* (40.38%), while *mcr-6* and *mcr-7* were not detected at all (Table S1).

The congruence of relative counts in isolates and relative abundance levels in metagenomes varied depending on which allele and what kind of sample grouping it

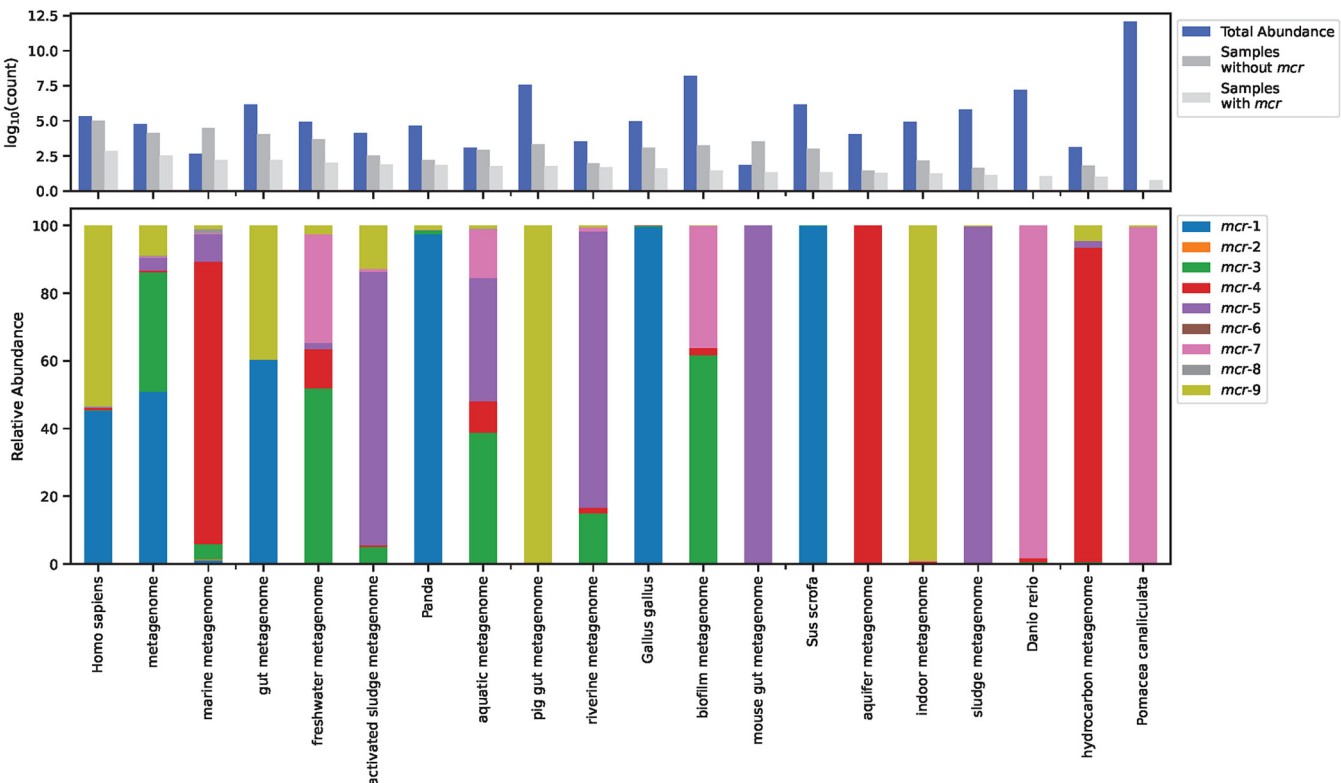

**FIG 3** Distribution of *mcr* genes in selected sampling hosts. Hosts were selected based on the host showing significant CLR values according to the ALDEx2 analysis (Fig. 4). (Top) Bar plot showing both the number of samples without and with an *mcr* hit and the overall *mcr* level for each host measured by log-abundance values. (Bottom) The abundance of individual *mcr* genes relative to total *mcr* levels. Data were normalized with gene lengths before plotting. To see the distribution of *mcr* genes for all sampling hosts available, refer to Fig. S2.

was. Grouped by the sampling location, the *mcr-1* gene appeared to be more widespread according to the isolates, whereas *mcr-3* had a larger global distribution based on the metagenomes. Similarly, for human samples, *mcr-1* had a higher prevalence in isolates, whereas metagenomes showed a higher abundance of *mcr-9*-aligned read fragments (Fig. S5).

**Genomic background of *mcr* genes.** The dissemination of colistin resistance genes between different reservoirs and countries in different sampling years was further investigated by creating assemblies of metagenomic samples with 95% coverage of at least one variant. We assembled 869 metagenomes, where we found 1,939 different contigs carrying *mcr* genes (range, 1 to 20 *mcr* contigs per metagenome). The most frequent gene present on these contigs was *mcr-9*, followed by *mcr-3* and *mcr-5* (Fig. S6a). To identify structural patterns between different metagenomic origins, we analyzed the genetic signatures in regions up- and downstream of an *mcr* gene (the flanks) with a minimum size of 1,000 bp and a maximum of 21,000 bp to include most of the elements found in the flanks (Fig. S6b). As most contigs were shorter than 1,000 bp (Fig. S6a), only 138 contigs passed the size criteria. All 20 contigs containing plasmid replicons in their flanks carried *mcr-1* genes, whereas the 63 flanks with MGEs were on contigs with different *mcr* variants (Fig. S6c).

Six distinct clusters became apparent upon calculating the distance between the flanks surrounding the *mcr* genes (Fig. 5). We find that the presence of specific MGEs seemingly correlated with the presence of an *mcr* variant on the contig, as IS*Apl1* occurs only on *mcr-1* contigs, and IS*903*, on *mcr-9* contigs. Five of the six clusters are all flanks around the same variant, with two being *mcr-1* clusters, while the sixth contains flanks surrounding four different *mcr* gene variants.

In the first *mcr*-1 cluster, an IncX4 plasmid replicon was present either upstream or downstream in most members. These contigs were found in metagenomes sampled in

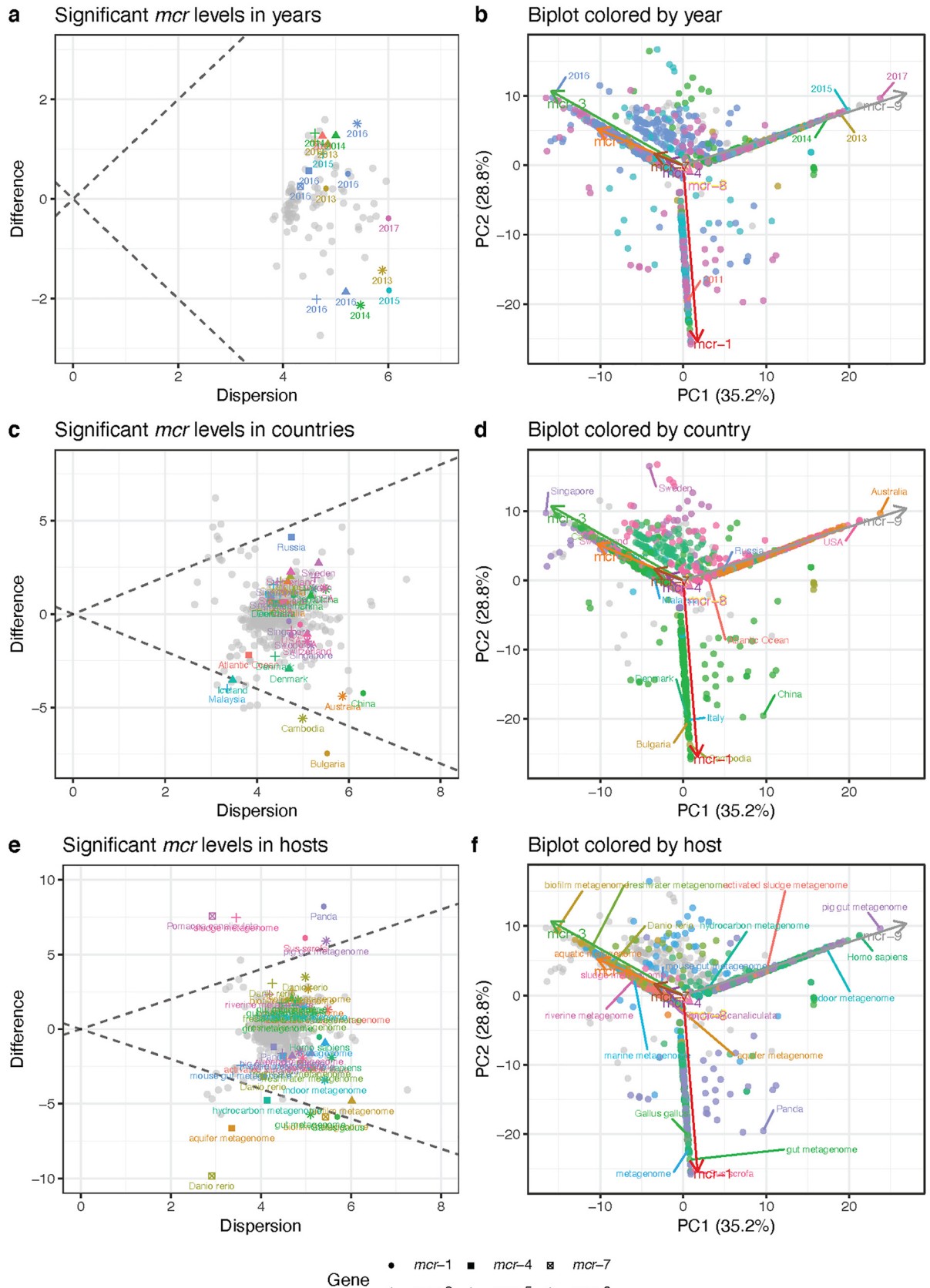

**FIG 4** Analysis of significant *mcr* levels in sampling years, countries, and hosts. (a, c, and e, left column) Visualizations of within-group dispersion of CLR values of individual *mcr* genes compared to the between-group difference in CLR values for (a) sampling

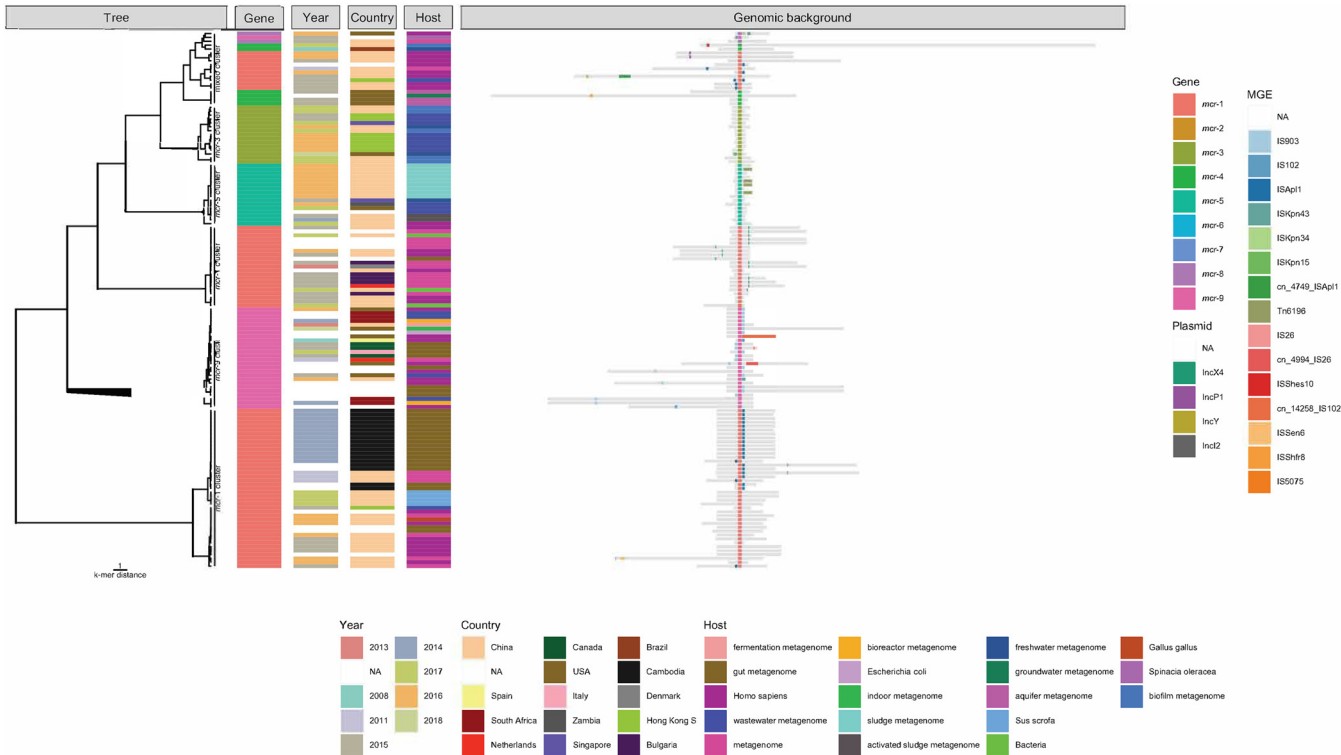

**FIG 5** Clustering of *mcr* contigs reveals that the genomic context remains conserved. The k-mer distance tree for flanks of *mcr* contigs with flank sizes between 1,000 bp and 21,000 bp is drawn in the left panel, with the metagenomic origins (year, country, host) added as colored tiles in the middle panel. The genomic background on the right is a schematic illustration of the size of each flank region in gray centered on the middle of the *mcr* gene and plasmid replicons and mobile genetic elements (MGEs) colored in the flanks.

2016 and 2017 from a diverse background, indicating that IncX4 plasmids are involved in multiple transmission events in different settings. The second *mcr-1* cluster differs from the first in that we see an absence of IncX4 and IncI2 in a few contigs instead. The cluster can instead be best characterized by the presence of the insertion element ISA*pl1* in half of the flanks, which mainly originate from gut metagenomes from Cambodia in 2014, while those without the insertion element are from Chinese samples from 2015 to 2017.

Four of the *mcr-9* contigs contain IS*102*, IS*26*, or IS*Kpn43*, while the remaining carry IS*903*. Since all these contigs contain at least one insertion sequence, it suggests that they were highly mobilized between different metagenomes from human and environmental sources between 2008 and 2018.

The mixed cluster shows a surprising clustering of flanks on contigs of uneven lengths, with different MGEs and replicons present, carrying four different *mcr* gene variants—*mcr-1*, *mcr-4*, *mcr-8*, and *mcr-9*. This indicates that despite the contigs carrying different *mcr* genes, there are similarities in their broader genomic context, despite it not being obvious how they are connected considering their various sample types.

**Metagenomic evidence of new *mcr* subvariants.** The varied origins of the collected metagenomes can be used to investigate how conserved known *mcr* gene variants are in different sources, as well as provide evidence of the presence of new *mcr* subvariants. Overall, most of the *mcr* reference sequences could be recovered from the metagenomic samples, although a large proportion seems only to be fragmented

**FIG 4** Legend (Continued)

year, (c) location, and (e) host. (b, d, and f, right column) Compositional biplots of the first two principal components (PC) capturing 64% of the variation in the data set, where samples are colored according to significant (b) years, (d) countries, and (f) hosts. Gray filled markers are samples that were nonsignificant. CLR: centered-log ratio transformed values of the proportion of *mcr* aligned read fragments.

**TABLE 2** Coverage of mcr templates according to KMA[a]

| Gene | No. of | | Template coverage (%) | | | | Depth of coverage (×) | | | |
|---|---|---|---|---|---|---|---|---|---|---|
| | Samples | Known subvariants | Avg. | Min. | Mdn. | Max. | Avg. | Min. | Mdn. | Max. |
| *mcr-1* | 418 | 12/14 | 71.238 | 1.05 | 85.980 | 100.18 | 43.877 | 0.01 | 2.350 | 1315.16 |
| *mcr-2* | 12 | 2/2 | 34.694 | 1.05 | 1.240 | 100 | 2.435 | 0.01 | 0.060 | 11.27 |
| *mcr-3* | 1,204 | 25/25 | 37.170 | 1.05 | 30.595 | 100.8 | 2.822 | 0.01 | 0.580 | 209.51 |
| *mcr-4* | 565 | 5/6 | 35.112 | 1.11 | 21.590 | 100 | 1.684 | 0.01 | 0.350 | 67.77 |
| *mcr-5* | 1,100 | 2/2 | 40.030 | 1.1 | 31.990 | 100 | 1.705 | 0.01 | 0.490 | 118.32 |
| *mcr-6* | 4 | 1/1 | 64.982 | 1.24 | 84.755 | 89.18 | 3.177 | 0.01 | 2.335 | 8.03 |
| *mcr-7* | 384 | 1/1 | 28.138 | 1.17 | 6.665 | 100.06 | 8.103 | 0.01 | 0.120 | 209.91 |
| *mcr-8* | 32 | 1/1 | 18.987 | 1.06 | 1.325 | 100 | 1.919 | 0.01 | 0.010 | 30.31 |
| *mcr-9* | 2,148 | 1/1 | 50.690 | 1.17 | 39.570 | 102.53 | 23.161 | 0.01 | 0.690 | 3,985.21 |

[a]The table contains an overview of the found number of *mcr* subvariants out of how many were known in the metagenomic samples, as well as summary statistics of template coverage and depth of coverages. Avg., average; Min., minimum; Mdn., median; Max., maximum.

sequences (Table 2). We constructed consensus sequences that had at least 90% template coverage, mean coverage depth of 5, and query identity of at least 90% and kept single nucleotide polymorphisms (SNPs) that had a minimum depth of 5 and 90% frequency. Of the 968 sequences constructed, 27.38% had at least one SNP difference in their template (Table 3). The majority of consensus sequences recovered from the metagenomes, whether they were known or new potential subvariants, could only be recovered in a few samples. Although, there are a few groups that stand out. We found 33 different subvariants of *mcr-3* genes, 32 of *mcr-7.1*, and a highly prevalent subvariant of *mcr-9.1* (Table 3, Fig. 6). Since these sequences were constructed from metagenomic samples with KMA, we call SNP variants for potential new subvariants.

The number of *mcr-3* subvariants in our version of the ResFinder database is 25, making it the variant with most subvariants. We found evidence of 33 new subvariants, though most only appear in a small number of samples, except for a subvariant of *mcr-3.6* called *mcr-3.6.v1* (Fig. S7a) and a subvariant of *mcr-3.15* called *mcr-3.15.v2* (Fig. S7b). Both *mcr-3.6.v1* and *mcr-3.15.v2* were detected in genomes of *Aeromonas* species (Table S2).

None of the *mcr-7.1* constructed sequences was an exact match to the reference sequence, and instead, we saw various numbers of SNPs (Fig. 6, Fig. S8a). While the frequencies of each new possible *mcr-7.1* variant in the metagenomes were not high, there appear to be several SNPs that were well conserved, for example, the two SNPs A1020G and A1275T present in 29 and 30 of the variants, respectively (Fig. S8a). Many of the 32 possible subvariants of *mcr-7.1* were found in water sources (e.g., zebrafish, freshwater, and wastewater) sampled over a period of 4 years (2016 to 2019) (Fig. 6). Unfortunately, none of the possible *mcr-7.1* subvariants had complete BLAST matches (Table S2).

We saw a high occurrence of a new subvariant sequence of *mcr-9.1*, named *mcr-9.1.v4*, which contained two SNPs, A1619G and A1620G (Fig. S8b). *mcr-9.1.v4* appears to originate in human or gut samples, a similar distribution to that of the template

**TABLE 3** Overview of SNP variant calling on consensus sequences

| Gene[a] | Total sequences[b] | SNP variants (%) | Unique SNP subvariant sequences[c] |
|---|---|---|---|
| *mcr-1* | 170 | 1.18 | 2 |
| *mcr-2* | 4 | 100.00 | 4 |
| *mcr-3* | 127 | 47.20 | 33 |
| *mcr-4* | 33 | 36.40 | 5 |
| *mcr-5* | 58 | 1.72 | 1 |
| *mcr-6* | 0 | 0 | 0 |
| *mcr-7* | 39 | 100.00 | 32 |
| *mcr-8* | 3 | 66.70 | 2 |
| *mcr-9* | 534 | 27.20 | 6 |
| Total | 968 | 27.38 | 62 |

[a]The number of consensus sequences per *mcr* gene.
[b]The percentage of consensus sequences found that are SNP variants.
[c]The number of unique SNP variant sequences recovered.

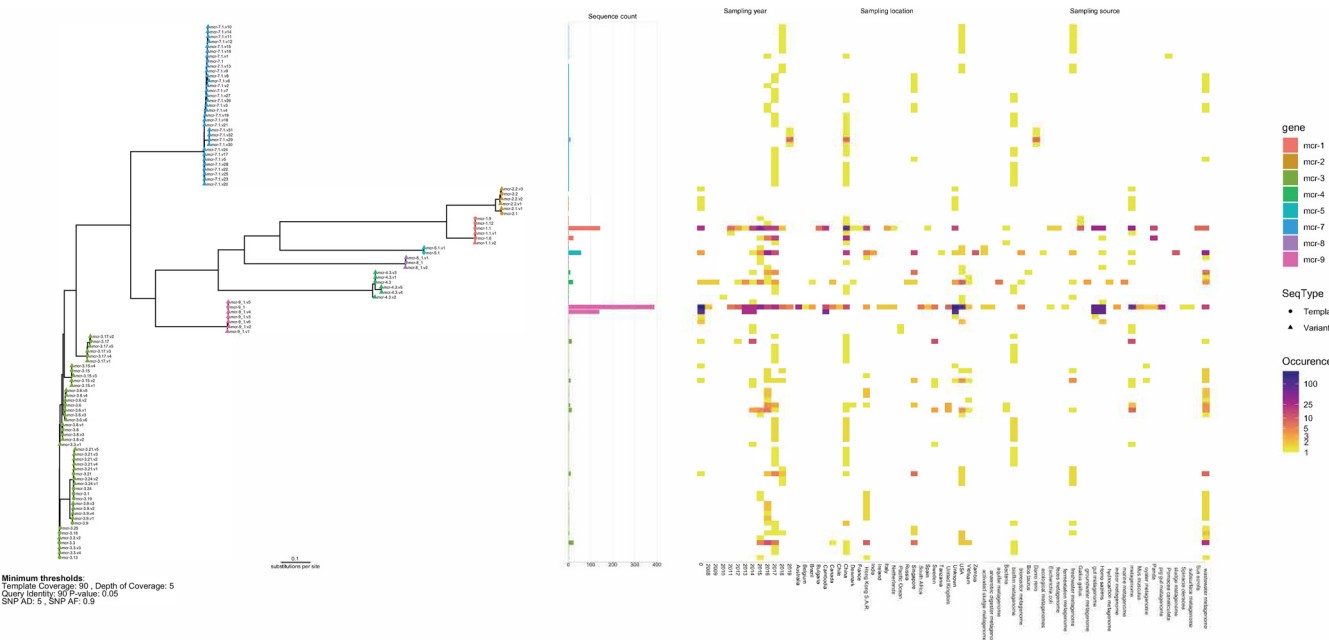

**FIG 6** Phylogenetic tree of consensus sequences. Sequences were aligned with MAFFT and clustered with FastTree. On the right is the occurrence of each sequence variant in different sampling origins (year, location, and source).

*mcr-9.1* (Fig. 6). *mcr-9.1.v4* had 20 BLAST hits, where the two most common species carrying this gene were *Enterobacter hormaechei* and *Salmonella enterica* subsp. *enterica* (Table S2).

## DISCUSSION

The growing collection of metagenomic sequencing data available in public repositories has the potential to provide a much more detailed picture of the emergence, evolution, and spread of ARGs. We downloaded and analyzed 214,095 host-derived and environmental metagenomes to characterize the global distribution of 9 mobilized colistin resistance genes that have been identified since 2015. Among the downloaded samples, we found that 4,465 metagenomes (2%) contained *mcr* reads distributed differently across the sampling period (2003 to 2019) and geographical and host origins. We found that all the nine different gene variants were present in metagenomes sampled years before their discovery (Fig. 1), confirming the notion of the resistance genes circulating in the environment long before being reported (18, 19, 23). This confirms the value of publicly sharing raw next-generation sequencing (NGS) data to promote new, better, and more comprehensive analyses of existing data.

To date, *mcr-1* is the most studied *mcr* gene variant, and the dissemination has been described in detail. Multiple studies agree that even though *mcr-1* has been detected in a few isolates from the 1980s (18), it has been appearing with increasing frequencies in samples between 2011 and 2017 and was decreasing in later years (8, 18, 19, 24). We see a similar trend in the frequencies of *mcr-1*-positive metagenomes, although the levels seem to increase starting from 2008 to the highest levels in 2015, the year of *mcr-1* first being reported (4) (Fig. 1).

In just a few years after discovering *mcr-1*, multiple other *mcr* variants were reported in different world regions, with *mcr-1* and *mcr-9* being the most disseminated genes (6). Despite *mcr-9* being the newest member (17), we observe that it was the most abundant gene variant in publicly available metagenomes, with *mcr-1* being the second most abundant. The two variants are not equally distributed across sampling sites, as *mcr-1* appears to be more geographically restricted to Europe and Asia, whereas *mcr-9* has reached a wider area (Fig. 2). In human metagenomes, *mcr-1* and *mcr-9* dominate, whereas other hosts and environmental origins display a considerable

variation in *mcr* variants. Despite earlier reports of the presence of *mcr-1* of both animal (4) and environmental (25) origins, we see only very few environmental origins of the gene (Fig. S2). Only a few hosts have significantly different levels of *mcr* gene abundances than expected, where *mcr-1* and *mcr-9* tend to be higher in pigs and pandas, and *mcr-3*, *mcr-4*, *mcr-5*, and *mcr-7* are lower in other hosts. *Mcr-2*, *mcr-6*, and *mcr-8* only appear in very few metagenomes, with *mcr-6* being the rarest variant. Since the first report of *mcr-6* (14), it has only been detected in very few places around the world and all in 2014 and 2015 (14, 26), but we can here report the presence of *mcr-6* in very small amounts in a metagenome from France sampled in 2012 (Fig. 1 and 2). Overall, there appears to be a connection between the abundance of a variant and the sampling source and location, but due to the sparse nature of our data set, we have not been able to determine the relative contribution of these factors to the observed *mcr* levels.

When observing the trends of the aligned *mcr* read fragment abundances in the data set, one should keep in mind that this collection is restrained by what was available in ENA at the time of download. The type of metagenomic data sets available is dependent on the ongoing research trends in the different scientific communities, which can cause a bias toward specific hosts or environments, such as the panda samples, by being overrepresented in the repository. Furthermore, there are challenges due to improved experimental protocols and sequencing platforms becoming available, possibly causing mapping bias. On the other hand, the evidence of the number of read fragments that match a specific gene should not be discarded too easily regardless of the sample origin. We applied compositional methods that can handle the nature of various read counts to ensure that the observed abundance levels of the different *mcr* alleles were not simply due to chance.

The NCBI Pathogen Detection Project is another example of a surveillance program that routinely screens available public data, in this case, genomes of single isolates. This data collection also has the same biases as those highlighted for our metagenomic collection, where our comparison of the two resources showed that each resource is in some cases better at capturing the prevalence of specific *mcr* alleles than the other. Essentially, our study highlights the benefit of using metagenomic data sets in addition to single isolates to monitor the distribution of AMR.

Interestingly, we observed that the *mcr* contigs from the assembled metagenomes were well conserved across reservoirs and locations except for *mcr-1* contigs. This suggests that most of the *mcr* alleles have only been mobilized once and then spread globally and between reservoirs. In contrast, *mcr-1* is known to be present in a variety of genomic backgrounds (8), which we also observe as the flanking regions of our *mcr-1* contigs grouped together in three distinct clusters (Fig. 5). *Mcr-1* is commonly found on IncI2, IncHI2, and IncX4 plasmids with ISA*pl1* (8, 25), although we only observed ISA*pl1* on two IncI2 plasmids, a possible loss of ISA*pl1* near IncX4 replicons, and we observed that no IncHi2 plasmids were present on *mcr-1* contigs. The absence of ISA*pl1* in one of the *mcr-1* clusters could indicate a loss of mobility due to either their difference in sampling years or a shift in hosts. IS*26* has been observed downstream of *mcr-9* (27), which we only observed once, and instead, we see that IS*903* occurs on both sides of *mcr-9* in the examined contigs. The metagenomic origin of *mcr-9* contigs is highly diverse, suggesting that the presence of multiple different insertion sequences has been a contributing factor in their mobilization between 2008 and 2018.

Even with the diverse genomic context of *mcr-1*, only very few of the *mcr-1* consensus sequences we constructed contained any SNPs, indicating that despite the different mobilization factors, the different *mcr-1* subvariants remain well conserved (Fig. 6). On the contrary, the sequences of *mcr-3* subvariants were highly prone to contain SNPs, as shown by our report of 33 potential new members, where several of them could be matched to genes in known species with BLAST (Table S2). Similarly, the diverse origin of *mcr-9* contigs is also reflected in the fact that an unknown subvariant

of *mcr-9.1*, which we are calling *mcr-9.1.v4*, was detected in 100 different genomes, with the species *Enterobacter hormaechei* being the most common (Table S2). We hesitate to call these SNP variants new variants, as more work needs to be done to test the expression levels and susceptibility of the organism carrying one of these potential subvariants, although there is strong evidence for the *mcr-9.1.v4* variant already being widely distributed.

As we collected data by downloading publicly available metagenomic samples, we present a data set with uneven coverage of sampling locations and sources. This bias heavily influences our ability to provide an in-depth understanding of the mobilization, emergence, and spread of the *mcr* genes. Regardless of this, we have shown the potential of using raw sequencing reads generated by other researchers to improve our knowledge. It is, however, important that all such generated data are shared publicly to allow for future exploration and improved understanding of the global microbial biology (28). Since the start of this project, another *mcr* variant was discovered named *mcr-10* (29), but we decided not to include the gene due to the massive computation task of mapping 442 Tb of raw sequencing reads. Nevertheless, it will be indeed interesting to figure out when *mcr-10* first appeared and characterize its dissemination as well, which we hope to do for this and for other ARGs in the future.

## MATERIALS AND METHODS

**Data collection.** Metagenomic data sets were collected from the public data repository the European Nucleotide Archive (ENA) (30). We queried the ENA API for samples uploaded between 1 January 2010 and 1 January 2020 that were shotgun sequenced and had at least 100,000 sequencing reads. In total, we downloaded 214,095 sequencing runs from 146,732 samples from 6,307 projects corresponding to 442 Tbp of raw reads.

**Reference sequence databases.** The AMR gene database ResFinder (31) (downloaded 25 January 2020) contains 3,085 sequences. The 16S rRNA SILVA (32) gene database (version 138, downloaded 16 January 2020) contains 2,225,272 sequences.

**Preprocessing and mapping sequencing reads.** The raw FASTQ reads were quality checked using FastQC v.0.11.15 (https://www.bioinformatics.babraham.ac.uk/projects/fastqc/) and trimmed with BBduk2 36.49 (33) to remove low-quality sequences and adaptors. BBduk2 settings were set as follows: minimum read length set to 50 bp, k=19, kmin=11, tbo flag on 11, the Phred quality threshold at 20 (99% accuracy), and only right trimming (ktrim=r). Assignment of trimmed reads to reference sequences was done with global alignment using KMA 1.2.21 (34) with the following alignment parameters: 1, -2, -3, -1 for a match, mismatch, gap opening, and gap extension. Also, a value of 7 for read pairing and a minimum relative alignment score of 0.75 were used. We used ResFinder to assess the number of acquired AMR reads in each sample and Silva to determine the bacterial content. On average, it took 5.7 s per metagenome for ResFinder mapping and 232.7 s for Silva mapping on a node equipped with dual 20 core Xeon Gold 6230 CPUs clocked at 2.1 Ghz using the Danish National Supercomputer for Life Sciences (https://www.computerome.dk).

**Compositional data analysis.** The collected metagenomic data have large variability in how the samples were collected, how DNA was extracted, and how it was sequenced. Furthermore, the probability of observing a gene also depends on the sequencing depth. To account for some of the variability, we use read fragment counts as the gene counts for mapping against ResFinder genes, and they were adjusted by individual gene lengths. Bacterial 16S read fragment counts from Silva mapping were aggregated to a total sum for each sample and divided by a million.

Abundance tables of *mcr* genes were created by transforming the composition $x$ of *mcr-1* to *mcr-9* length-adjusted counts $n_i$ ($i = 1 \ldots 9$) and the summed per million bacterial component $n_B$ by using the bacterial component as the reference and log-transforming the ratios:

$$x = [n_1, n_2, \ldots, n_9, n_B]$$

$$\text{Abundance}(x) = \left( \log \frac{n_1}{n_B}, \log \frac{n_2}{n_B}, \ldots, \log \frac{n_9}{n_B} \right)$$

For the statistical analysis performed on the mapping results, we treated the *mcr* read fragment counts as compositional. If we do not consider the observed counts as being relative to each sample, statistical tests can produce faulty results. Instead, if we apply the methods of compositional data analysis, this is avoided. As proposed by Aitchison (35), we log-ratio transform the counts to make the data symmetric, linear, and in a log-ratio coordinate space.

However, before applying log-transformations, counts of zero needed to be treated. Since a zero does not necessarily mean that a gene is absent from a sample and the logarithm of zero is an undefined value, we infer the proportion $p_i$ of reads of an ARG $i$ within a sequenced sample directly from the observed read count $n_i$. If we assume that each $n_i$ was sampled from a Poisson process, $n_i \sim \text{Poisson}(\lambda_i)$,

and the vector of counts follows a multinomial distribution $\{[n_1, n_2, \ldots] \mid n\} \sim \text{Multinomial}(p_1, p_2, \ldots \mid n)$, where $n = \sum_i n_i$ and $p_i = \dfrac{\lambda_i}{\sum_k \lambda_k}$. The posterior distribution of $[p_1, p_1, \ldots$ is given as the product of the multinomial likelihood with a Dirichlet$\left(\frac{1}{2}, \frac{1}{2}, \ldots\right)$ prior. These inferred proportions will never be precisely zero, even if the observed count is zero because of the multivariate distribution (36).

We used the centered log-ratio (CLR) transformation on the zero-replaced composition consisting of *mcr* read proportions *p*, excluding the bacterial component:

$$p = [p_1, p_2, \ldots, p_9]$$

$$\text{CLR}(p) = \left( \log \frac{p_1}{g_m(p)}, \ldots, \log \frac{p_9}{g_m(p)} \right)$$

where $g_m(p) = \left( \prod_{i=1}^{D} p_i \right)^{\frac{1}{D}}$, $D = 9$ is the geometric mean of the composition. The CLR values were used as the input for differential abundance tests and principal-component analysis, as described below.

**Data visualization.** Graphics visualizing abundance and relative abundances were created with Python 3.8 with Matplotlib 3.3.2 (37) and seaborn 0.11.0 (38). Bar plots showing relative abundances were created by closing the composition to 100. Geographical maps showing gene abundances were created using Shapely 1 3166.7.1 (39) and Cartopy 0.18.0 (40) to translate labels from the metadata into geographical shapes with the Natural Earth data set.

**Statistical analysis.** We carried out a differential abundance analysis on samples containing *mcr* fragments with ALDEx2 (41) 1.18.0 in R. We aimed to identify which experimental groups showed a difference in their abundance of *mcr* gene read fragments compared to other groups. ALDEx2 tests for significant differences of CLR abundance between categorical sample groups used Welch's *t* test followed by a Benjamini-Hochberg false-discovery rate (FDR) correction (42). We report significant groups of either sample locations, host, or collection year where the FDR is <0.05 and differential abundances were represented in an effect plot (43) displaying the within- and between-group variation in CLR values.

Principal-component analysis (PCA) was applied to the centered, scaled by total variance, and CLR transformed data set of *mcr* read proportions (44) to reduce the dimensionality of the data. The eigenvectors and eigenvalues from PCA were used to create a biplot, highlighting the significant sample groups found in the differential abundance analysis.

**Comparison of metagenomic abundance levels to prevalence in pathogen isolates.** The NCBI Pathogen Detection Project (45) routinely screens new isolates to identify AMR genes with the tool AMRFinderPlus (46), which reports whether a gene was found or not in an assembled genome. We downloaded the annotation results of 912,469 assembled genomes from NCBI's Pathogen Detection Resource (https://ftp.ncbi.nlm.nih.gov/pathogen/Results/, accessed 5August 2021); 7,934 (0.87%) of the single isolates contained at least 1 of the 9 *mcr* variants. We reported the frequency of the number of isolates carrying each *mcr* variant. Furthermore, we grouped the isolates by either sampling year, location, or host and reported the relative count of each variant to the relative abundance levels in the metagenomes.

**Metagenomic assembly of *mcr* samples.** We assembled metagenomes where at least one of the *mcr* genes had a minimum coverage of 95% by trimmed reads, according to KMA. The trimmed reads were assembled with MetaSPAdes 3.14.0 (47) with at least 1.2 terabytes of memory, 40 threads per node, and a maximum runtime of 1 week. Out of 1,014 metagenomes, 145 were not assembled, as they did not complete within the chosen time frame of a week. Contigs carrying the nine different *mcr* gene variants were identified with blastn 11.0 (48) with a percentage identity of ≥95.

**Flank analysis of metagenomic assemblies.** The metagenomic contigs carrying *mcr* genes were used in the flank analysis. Flanks were created by masking the *mcr* gene in the contig and cutting out up- and downstream regions of increasing sizes between 1,000 bp and 30,000 bp by intervals of 1,000 bp with BEDTools (49). The presence of plasmids in the flanks was identified with PlasmidFinder 2.1 (50), and mobile elements, with MobileElementFinder 1.0.3 (51). The distance between the flanks was calculated as the Szymkiewicz-Simpson dissimilarity with KMA (34). Hierarchical clustering on the flank distances was done with Ward's method (52) to create a dendrogram plotted with ggtree 2.0.4 (53) and ggplot 3.3.3 (54). This approach is similar to the workflow of the tool Flanker by Matlock et al. (55), except that we cluster with KMA.

**Variant analysis of *mcr* genes.** We investigated the presence of SNPs in KMA-produced consensus sequences that matched the following minimum requirements: template coverage, ≥98%; depth of coverage, ≥5; query identity, ≥90%; and *P* value, ≤0.05. SNPs were kept if they passed the following filters, checked with bcftools 1.13 (56): a minimum allele depth of 5 (AD) and a minimum allele frequency of 0.90 (AF). Sequences were aligned with MAFFT v7.490 (57), and phylogenetic trees were created with FastTree v2.1.1 (58) using a nucleotide substitution model. Trees were visualized with ggtree. A visual summary of SNPs in sequence alignments was created with snipit (https://github.com/aineniamh/snipit).

We screened all unique sequences that had at least 1 SNP difference to their template against complete and draft genome sequences in GenBank with BLAST (https://blast.ncbi.nlm.nih.gov/Blast.cgi, accessed 24 January 2022). Matches were identified if they had 100% identity to the template.

**Data availability.** Source data for generating abundance figures and running statistical tests and flank and variant analysis can be found in the supplementary files at https://doi.org/10.5281/zenodo .5946866, and the supporting code is available at https://github.com/hmmartiny/mcr_metagenomes.

## SUPPLEMENTAL MATERIAL

Supplemental material is available online only.

**FIG S1**, TIF file, 2.9 MB.
**FIG S2**, TIF file, 0.6 MB.
**FIG S3**, TIF file, 0.3 MB.
**FIG S4**, TIF file, 0.6 MB.
**FIG S5**, TIF file, 2.6 MB.
**FIG S6**, TIF file, 0.7 MB.
**FIG S7**, TIF file, 0.4 MB.
**FIG S8**, TIF file, 0.7 MB.
**TABLE S1**, DOCX file, 0.01 MB.
**TABLE S2**, DOCX file, 0.02 MB.

## ACKNOWLEDGMENTS

This work was supported by the European Union's Horizon H2020 grant VEO (874735) and the Novo Nordisk Foundation (grant NNF16OC0021856: Global Surveillance of Antimicrobial Resistance).

F.M.A. and T.N.P. designed the project. H.-M.M. performed data acquisition, assembly, and analysis. J.S. and T.N.P. provided guidance with sequence downloading and mapping, P.M. and C.B. provided guidance with compositional analysis, P.M. provided guidance with flank analysis, and F.M.A., T.N.P., and P.M. provided guidance with SNP variant analysis. H.-M.M. wrote the draft. All authors have read, commented on, and accepted the final version.

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
