## [Reviewer comments · mSystems]

Global distribution of *mcr* gene variants in 214K metagenomic samples

Hannah-Marie Martiny, Patrick Munk, Christian Brinch, Judit Szarvas, Frank Aarestrup, and Thomas Petersen

Corresponding Author(s): Hannah-Marie Martiny, Technical University of Denmark

Review Timeline:

Submission Date:

February 3, 2022

Accepted:

February 28, 2022

Editor: Zackery Bulman

Reviewer(s): Disclosure of reviewer identity is with reference to reviewer comments included in decision letter(s). The following individuals involved in review of your submission have agreed to reveal their identity: Val Fernández-Lanza (Reviewer #1)

Transaction Report:

DOI: <https://doi.org/10.1128/msystems.00105-22>

February 28, 2022

Dr. Hannah-Marie Martiny
Technical University of Denmark
Kongens Lyngby
Denmark

Re: mSystems00105-22 (Global distribution of *mcr* gene variants in 214K metagenomic samples)

Dear Dr. Hannah-Marie Martiny:

Thank you for resubmitting your article to mSystems. Your manuscript has been accepted, and I am forwarding it to the ASM Journals Department for publication. For your reference, ASM Journals' address is given below. Before it can be scheduled for publication, your manuscript will be checked by the mSystems production staff to make sure that all elements meet the technical requirements for publication. They will contact you if anything needs to be revised before copyediting and production can begin. Otherwise, you will be notified when your proofs are ready to be viewed.

Publication Fees:

We recognize that the video files can become quite large, and so to avoid quality loss ASM suggests sending the video file via <https://www.wetransfer.com/>. When you have a final version of the video and the still ready to share, please send it to mSystems staff at mssystemsjournal@msubmit.net.

For mSystems research articles, if you would like to submit an image for consideration as the Featured Image for an issue, please contact mSystems staff at mssystemsjournal@msubmit.net.

Sincerely,

Zackery Bulman
Editor, mSystems

Journals Department
Figure S8: Accept
Table S2: Accept
Figure S7: Accept
Figure S1: Accept
Table S1: Accept
Figure S3: Accept
Figure S4: Accept
Figure S2: Accept
Figure S5: Accept
Figure S6: Accept